# Functional Selectivity of Coumarin Derivates Acting via GPR55 in Neuroinflammation

**DOI:** 10.3390/ijms23020959

**Published:** 2022-01-16

**Authors:** Matthias Apweiler, Jana Streyczek, Soraya Wilke Saliba, Juan Antonio Collado, Thomas Hurrle, Simone Gräßle, Eduardo Muñoz, Claus Normann, Sabine Hellwig, Stefan Bräse, Bernd L. Fiebich

**Affiliations:** 1Neuroimmunology and Neurochemistry Research Group, Department of Psychiatry and Psychotherapy, Medical Center-University of Freiburg, Faculty of Medicine, University of Freiburg, D-79104 Freiburg, Germany; matthias.apweiler@uniklinik-freiburg.de (M.A.); jana.streyczek@uniklinik-freiburg.de (J.S.); wilkesaliba@yahoo.com.br (S.W.S.); 2Departamento de Biología Celular, Fisiología e Inmunología, Universidad de Córdoba, E-14071 Cordova, Spain; md2coroj@uco.es (J.A.C.); fi1muble@uco.es (E.M.); 3Institute of Organic Chemistry, Karlsruhe Institute of Technology (KIT), D-76131 Karlsruhe, Germany; hurrle.thomas@gmail.com (T.H.); stefan.braese@kit.edu (S.B.); 4Institute of Biological and Chemical Systems-Functional Molecular Systems (IBCS-FMS), Karlsruhe Institute of Technology (KIT), Hermann-von-Helmholtz-Platz 1, D-76344 Eggenstein-Leopoldshafen, Germany; simone.graessle@kit.edu; 5Instituto Maimónides de Investigación Biomédica de Córdoba, E-14004 Cordova, Spain; 6Hospital Universitario Reina Sofía, E-14004 Cordova, Spain; 7Department of Psychiatry and Psychotherapy, Medical Center-University of Freiburg, Faculty of Medicine, University of Freiburg, D-79104 Freiburg, Germany; claus.normann@uniklinik-freiburg.de (C.N.); sabine.hellwig@uniklinik-freiburg.de (S.H.)

**Keywords:** neuroinflammation, GPR55, coumarin derivates, PGE_2_, functional selectivity, SK-N-SH, primary microglia

## Abstract

Anti-neuroinflammatory treatment has gained importance in the search for pharmacological treatments of different neurological and psychiatric diseases, such as depression, schizophrenia, Parkinson’s disease, and Alzheimer’s disease. Clinical studies demonstrate a reduction of the mentioned diseases’ symptoms after the administration of anti-inflammatory drugs. Novel coumarin derivates have been shown to elicit anti-neuroinflammatory effects via G-protein coupled receptor GPR55, with possibly reduced side-effects compared to the known anti-inflammatory drugs. In this study, we, therefore, evaluated the anti-inflammatory capacities of the two novel coumarin-based compounds, KIT C and KIT H, in human neuroblastoma cells and primary murine microglia. Both compounds reduced PGE_2_-concentrations likely via the inhibition of COX-2 synthesis in SK-N-SH cells but only KIT C decreased PGE_2_-levels in primary microglia. The examination of other pro- and anti-inflammatory parameters showed varying effects of both compounds. Therefore, the differences in the effects of KIT C and KIT H might be explained by functional selectivity as well as tissue- or cell-dependent expression and signal pathways coupled to GPR55. Understanding the role of chemical residues in functional selectivity and specific cell- and tissue-targeting might open new therapeutic options in pharmacological drug development and might improve the treatment of the mentioned diseases by intervening in an early step of their pathogenesis.

## 1. Introduction

A growing body of research demonstrates the enormous role of neuroinflammation in neurological and psychiatric diseases, such as Alzheimer’s Disease (AD), Parkinson’s Disease (PD), schizophrenia, and depression [1,2,3,4]. Therefore, pharmacological mechanisms targeting neuroinflammation in the treatment of the mentioned diseases might open new therapeutical options. A meta-analysis, covering 36 randomized clinical trials (RCTs) and a total of about 10,000 patients, showed improvement of depressive symptoms after an intervention with specific anti-inflammatory substances, such as non-steroidal anti-inflammatory drugs (NSAIDs) and glucocorticoids, as monotherapy or added to a classic pharmaceutical anti-depressive therapy compared to placebo [5]. For AD, the effects of anti-neuroinflammatory treatment on the progress of neurodegeneration are discussed controversially due to differing results between epidemiological and clinical studies. Nevertheless, new approaches based on statistical modelling using the Alzheimer’s Disease Neuroimaging Initiative showed better cognitive baselines after treatment with NSAIDs and especially diclofenac treatment reduced cognitive decline [6]. PD is associated with neuroinflammation as well, and for this reason, anti-inflammatory treatment with NSAIDs decelerate the disease’s progression and protects remaining dopaminergic neurons [7]. However, these drugs are associated with severe side-effects in their established indications, limiting the period and dose of treatment. A total of 10% of hospitalizations of elderly people are related to preventable drug side-effects, with NSAIDs being responsible for 30% of these avoidable drug reactions. A total of 6% of the patients taking NSAIDs experience side-effects with the need for a consultation of a general doctor within 2 months [8]. Glucocorticoids, for example, impair mitochondrial functions and therefore might foster neurodegeneration due to increased oxidative stress [9]. Therefore, novel anti-inflammatory compounds with fewer potential harmful side-effects might promote the use of anti-inflammatory drugs in diseases such as AD, PD, and depression. The underlying mechanisms of the positive effects of NSAIDs on the mentioned diseases, such as inhibition of mitochondrial Ca^2+^ overload beside the known cyclooxygenase (COX) inhibition, are still objects of ongoing studies [7,10]. The COX enzymes are mediating the enzymatic conversion of arachidonic acid (AA) to prostaglandin (PG) H_2_, which is in turn metabolized to PGE_2_ by different prostaglandin E synthases (PGES), such as microsomal (m)PGES-1 or mPGES-2 [11]. PGE_2_ is known to act as pro-inflammatory in tissues and the CNS and increased levels are linked to psychiatric disorders such as psychosis [12], and therefore its reduction by NSAID treatment might be beneficial in neuropsychiatric diseases [5] as mentioned above.

Coumarin-based derivates have been shown to elicit anti-neuroinflammatory effects in primary microglia, decreasing lipopolysaccharide (LPS)-induced PGE_2_ release [13,14]. High levels of PGE_2_ promote brain injury and therefore act as a pro-inflammatory driver [15]. It has been suggested that the effects of coumarin derivates rely on antagonism with inverse agonistic activity at the G-protein coupled receptor 55 (GPR55) [13,14,16]. A recent study evaluating the effects of the two coumarin derivates KIT C and KIT H, which are investigated in this current study as well, showed anti-oxidative effects of both compounds, which were abolished after GPR55 knockout in SK-N-SH cells [17].

The GPR55 was discovered and cloned in 1999 [18] and is broadly expressed in the CNS, especially in the frontal cortex, putamen, striatum, and caudate [19]. Different studies identified numerous exogenous as well as endogenous ligands of the receptor. Besides ligands known to interact with receptors of the endocannabinoid system, such as delta9-tetrahydrocannabinol (Δ9-THC), L-α-lysophosphatidylinositol (LPI) has been demonstrated to have a strong affinity to GPR55 and is therefore suggested as an endogenous ligand of GPR55 [20]. Besides the endogenous ligands, synthetic selective GPR55 ligands, such as O-1602, acting as an agonist, and the antagonistic ML 193, have been introduced and are commonly used in GPR55 studies, among others [19]. Furthermore, derivates of the coumarin scaffold have been synthesized and suggested to act as antagonists on GPR55 [16,21]. Our group demonstrated the anti-neuroinflammatory effects of different coumarin-based compounds with binding affinities to GPR55 named KIT 10 [14], KIT 3, KIT 17, and KIT 21 [13], showing anti-neuroinflammatory properties in primary microglial cell cultures.

The GPR55-mediated biological effects remain the focus of current research. However, GPR55 agonism might be associated with negative effects for the cells, tissues, and, consequently, the organism itself. Agonism and overexpression of GPR55 are associated with cancer proliferation [22], metabolic diseases, such as obesity and diabetes [23], and decreased osteoclast formation [24]. Therefore, GPR55 antagonists might reverse negative GPR55-mediated effects and open new therapeutical options in the treatment of several diseases. Various in vivo and in vitro studies with central nervous cells or tissues and model organisms are focusing on the effects of GPR55 expression and antagonists in different conditions and diseases. A model for AD, 5xFAD-mice, showed a higher expression of GPR55 in the hippocampus compared to heterozygotic and wildtype mice with impairments in novel object recognition [25]. In the chemically-induced murine PD model, chronic abnormal cannabidiol (GPR55 agonist) treatment improved motoric functions and acted neuroprotectively [26]. GPR55 agonists, as well as antagonists, enfolded beneficial effects on motor coordination and sensorimotor deficits on 6-hydroxydopamine-induced PD symptoms in rats [27], suggesting a more complex role of GPR55 in PD. Furthermore, intrahippocampal administration of the GPR55 agonist O-1602 protected against LPS-induced inflammatory insults of neural stem cells [28]. In another study, intracerebroventricular injection of O-1602 induced anxiolytic effects in an elevated plus-maze test in rats, whereas ML 193 led to increased anxiety-like behavior [29]. In the corticosterone-induced depressive-like behavior of rats, O-1602 reversed depressive-like behavior and normalized increased levels of interleukin (IL)-1β and tumor necrosis factor (TNF)α [30]. GPR55-knockout mice repealed hyperalgesia to mechanical stimuli suggesting GPR55 to be a promising target for treating inflammatory and neuropathic pain [31]. The featured studies indicate a complex role of GPR55 in neurological and psychiatric diseases, with the agonism, as well as the antagonism, being beneficial dependent on the concrete situation. Association of a GPR55-mutation with psychiatric diseases has been shown in human clinical trials as well. In suicide victims without any diagnosed mental illness, decreased GPR55 and CB2 gene expression with increased GPR55-CB2 heteromers were found in the dorsolateral prefrontal cortex (DLPFC), eliciting a potential involvement of GPR55 in impulsivity and decision-making in suicide [32]. The single nucleotide polymorphism Gly195Val of the GPR55 is associated with an increased risk of Anorexia nervosa (AN) in a study comparing Japanese AN-patients with an age-unmatched control group [33].

GPR55 transduces extracellular signals via Gα_12/13_ [34] and Gα_q_ [35], resulting in the phosphorylation and activation of phospholipase C, protein kinase C (PKC), mitogen-activated protein kinases (MAPK) such as p38 MAPK, and extracellular signal-regulated kinase (ERK), followed by the activation of transcriptional factors [19]. The activation of the different pathways is complexly regulated and might differ between various ligands [36]. These phenomena might be explained by different primary active states of one receptor as a response to different ligands resulting in distinct conformations responsible for the selective pathway activation, also referred to as functional selectivity [37,38]. As shown in a previous study [13], comparing the three coumarin-based compounds, KIT 3, KIT 17, and KIT 21, the effects on PGE_2_-reduction of the compounds show enormous differences probably dependent on the chemical residues, which might be explained by functional selectivity for antagonists. In contrast to GPR55 agonists, GPR55 antagonists are defined by a head region with the most electronegative region near the end of the central portion, whereas agonists have the electronegativity in the head region. Furthermore, GPR55 antagonists show an aromatic or heterocyclic ring that protrudes out of the binding pocket of GPR55, potentially preventing any conformational change [19]. Therefore, the different residues of the tested coumarin-based compounds [13,14] might determine how deep the compounds fit in the binding pocket and therefore, how potent the compounds might change the receptors state following the extent of the biological effects.

For this current study, two novel synthesized coumarin derivates named KIT C and KIT H were investigated in human neuroblastoma cells and in primary microglial cell cultures of mice. The effects of KIT C and KIT H on the COX-2/PGE_2_ pathway and pro- and anti-inflammatory mediators were evaluated in comparison to the commercial GPR55 agonist O-1602 and antagonist ML 193.

## 2. Results

### 2.1. Effects of the Compounds on Cell Viability

Results of the performed MTT cell viability assay for the used compounds are presented in Figure 1. Neither KIT C (light grey bars) or KIT H (dark grey bars), as shown before [17], nor O-1602 (light blue bar) or ML 193 (blue bar) showed cytotoxic effects in IL-1β-stimulated SK-N-SH cells compared to untreated cells. KIT C in concentrations of 5 and 10 µM, 1 µM KIT H, and 25 µM ML 193, on the contrary, significantly increased cell viability or metabolism. Ethanol, used as positive control, strongly induced cell death as expected. Since none of the compounds in the concentrations tested elicited cytotoxic effects, we proceeded with further experiments.

### 2.2. Effects of the Compounds on IL-1β-Induced PGE_2_-Release

Since PGE_2_ is the central molecule in the AA/COX-2/PGE_2_ pathway and acts pro-inflammatory, next we investigated the effects of KIT C, KIT H, O-1602, and ML 193 on PGE_2_-release in IL-1β-stimulated SK-N-SH cells. KIT C (light grey bars), as well as KIT H (dark grey bars), showed a significant and concentration-dependent reduction of IL-1β-induced PGE_2_-levels (Figure 2) starting at concentrations of 5 µM. KIT H elicited a more potent PGE_2_-reduction than KIT C, reaching basal PGE_2_-concentrations of untreated cells in the concentration of 25 µM. ML 193 (blue bar), a GPR55 antagonist, also showed significant inhibition of IL-1β-mediated PGE_2_-release, with an effect size between KIT C and KIT H. O-1602 (light blue bar), a potent GPR55 agonist, did not significantly inhibit IL-1β-induced PGE_2_-synthesis.

### 2.3. GPR55 Activity of KIT C and KIT H

To prove whether the observed anti-inflammatory effects of KIT C and KIT H are mediated via GPR55, a GPR55 activation assay was performed (Figure 3). AM251 (1 µM), a GPR55 agonist with additional activities at CB1- and CB2-receptors, and LPI (10 µM), the physiological agonist of GPR55, were used as positive controls. KIT C in concentrations of 5 and 10 µM showed about a 4-fold higher GPR55 activation than 1 µM AM251 without reaching significance but showing a clear trend. KIT H revealed about a 2-fold but not significantly higher GPR55 activation than 1 µM AM251 in all tested concentrations, comparable to the GPR55 activation capacity of 10 µM LPI.

### 2.4. Effects of KIT C and KIT H on COX-2 mRNA and Protein Levels

To investigate the underlying mechanisms of the strong PGE_2_-reduction, COX-2 expression and synthesis were evaluated using Western Blot (Figure 4A) and qPCR (Figure 4B). COX-2 protein synthesis was potently increased by IL-1β if compared to the untreated control. Pre-treatment with exclusively 5 µM KIT C (light grey bars) reduced IL-1β-stimulated COX-2 levels in SK-N-SH cells. KIT H (dark grey bars) significantly reduced IL-1β-mediated COX-2 synthesis starting with the concentration of 1 µM. As shown in Figure 4B, COX-2 mRNA expression was potently induced by IL-1β-stimulation for 4 h. Whereas KIT H (dark grey bars) did not affect IL-1β-induced COX-2 expression, KIT C (light grey bars) significantly enhanced IL-1β-induced COX-2 expression in concentrations of 0.1, 5, and 10 µM which contrasts with the Western Blot results. Treatment with KIT C for different time points (2, 4, 8, 12, 24 h) followed by the analysis of COX-2 protein synthesis and mRNA expression did not explain the observed diverging effects on COX-2 synthesis and expression (Appendix A), so we can exclude effects based on differences in the IL-1β-stimulation time course. COX-2 mRNA expression was higher or at least comparable to the IL-1β positive control at all time points, whereas COX-2 protein levels were first detectable after 8 h at higher levels than IL-1β-stimulated cells and started to decrease after 12 h of stimulation. After 24 h, COX-2 expression remained comparable to or higher than IL-1β-treated cells, while protein levels were significantly reduced as shown in Figure 4.

### 2.5. Effects of KIT C and KIT H on COX-Activity

Besides COX synthesis and expression, enzyme activities of COX-1 (Figure 5A) and COX-2 (Figure 5B) were examined as another possible mechanism of PGE_2_-reduction independent of COX synthesis and expression. Neither KIT C (light grey bars) nor KIT H (dark grey bars), the GPR55 agonist O-1602, and antagonist ML 193 affected COX-1 or COX-2 activities in concentrations between 0.1 and 10 µM. Both COX inhibitor controls potently decreased COX activities. The selective COX-1 inhibitor SC-560 decreased COX-1 activity by about 70% but did not reach significance. The COX-1 and COX-2 inhibitor diclofenac in concentrations of 0.1 and 1 µM significantly reduced COX-2 activity.

### 2.6. Effects of KIT C and KIT H on COX-1 and mPGES-1 Expression

Next, we studied the effects of KIT C (light grey bars) and KIT H (dark grey bars) on two other important enzymes involved in the AA/PGE_2_ pathway, COX-1 and mPGES-1. The expression of both enzymes was evaluated using qPCR. The expression of mPGES-1 (Figure 6A) was strongly induced by IL-1β-treatment for 4 h and 10 µM of KIT C slightly but significantly increased mPGES-1 expression compared to the IL-1β positive control. KIT H did not affect IL-1β-stimulated mPGES-1 expression. COX-1 expression (Figure 6B) was decreased by stimulation with IL-1β, and KIT C, as well as KIT H, partially ameliorated the IL-1β-induced reduction of COX-1 expression.

### 2.7. Effects of KIT C and KIT H on IL-1β–Induced Cytokine Release

Besides the AA/PGE_2_ pathway, the effects of KIT C (light grey bars) and KIT H (dark grey bars) on IL-1β-induced IL-6 as pro-inflammatory and IL-10 as anti-inflammatory cytokines were investigated (Figure 7). Stimulation with IL-1β for 24 h potently induced IL-6 release in SK-N-SH cells but neither KIT C nor KIT H nor O-1602 affected IL-6 production (Figure 7A). The GPR55 antagonist ML 193, however, significantly reduced IL-1β and increased IL-6 release by about 60% in IL-1β-treated SK-N-SH cells.

IL-1β reliably induced IL-10 mRNA expression in SK-N-SH cells as shown by qPCR, and KIT C as well as KIT H both enhanced IL-1β-stimulated IL-10 mRNA levels in concentrations of 10 µM compared to IL-1β the positive control (Figure 7B).

### 2.8. Effects of KIT C and KIT H on PGE_2_- and IL-6 Release in LPS-Stimulated Primary Mouse Microglia

The promising results of KIT C and KIT H in SK-N-SH cells were re-evaluated in primary mouse microglia as preliminary results for follow-up studies. Only KIT C (light grey bars) but not KIT H (dark grey bars) significantly reduced PGE_2_-levels after LPS-induction in primary microglia (Figure 8A). A total of 10 µM KIT C reduced PGE_2_ levels to concentrations compared to untreated primary microglia.

LPS-stimulation for 24 h potently induced IL-6 production (Figure 8B) in primary mice microglia as well. KIT C (light grey bars) significantly decreased LPS-induced IL-6-synthesis by about 50% and KIT H (dark grey bars) showed a non-significant trend of reducing IL-6 release in primary mice microglial cells.

## 3. Discussion

The current study investigates the anti-neuroinflammatory effects of two novel coumarin derivates, KIT C and KIT H, in SK-N-SH cells as well as in primary microglial cell cultures. Selected results of KIT C and KIT H were compared to the commercial GPR55 agonist O-1602 and the GPR55 antagonist ML 193. None of the tested compounds showed toxic effects in SK-N-SH cells. KIT H demonstrated the most potent reduction of IL-1β-induced PGE_2_-levels compared to KIT C and ML 193, whereas O-1602 did not affect PGE_2_-levels. In primary mice microglia, KIT C decreased LPS-induced PGE_2_-release, whereas KIT H did not affect PGE_2_-levels significantly. Although KIT C increased IL-1β-induced COX-2 expression in SK-N-SH cells, both coumarin derivates decreased COX-2 protein synthesis. Neither COX-1 nor COX-2 enzymatic activity was affected by the tested compounds and thus, the PGE_2_ decreasing effects are not due to direct inhibition of those enzymes as shown for classical NSAIDs. Furthermore, the IL-1β-mediated reduction of COX-1 expression was partially reversed by KIT C and KIT H. In SK-N-SH cells, ML 193 significantly decreased IL-1β induced IL-6 levels, but neither KIT C nor KIT H or O-1602 affected IL-6 release. In primary microglia, however, KIT C but not KIT H significantly inhibited LPS-induced IL-6 release. IL-1β-mediated IL-10 expression was significantly increased by KIT C and KIT H in SK-N-SH cells.

The anti-neuroinflammatory effects of other coumarin derivates have been shown using different coumarin derivates (KITs) in primary rat microglia, potently inhibiting LPS-induced PGE_2_-release [13,14]. Our group showed a strong reduction of PGE_2_-synthesis, COX-2 gene expression, and protein synthesis as well as mPGES-1 protein synthesis after treatment with KIT 17 [13]. KIT 10 also reduced COX-2 and mPGES-1 protein synthesis [14]. Neither KIT 17 nor KIT 10 affected COX-2 enzymatic activity, whereas KIT 17 but not KIT 10 increased COX-1 activity [13,14]. The demonstrated reduction of COX-2 protein synthesis by KIT 17 was replicated in the current study for KIT C and KIT H. Since COX-2 is a key step in the enzymatic transformation of arachidonic acid to PGH_2_ [39], the reduction of COX-2 protein levels might be at least partially responsible for the observed PGE_2_-reduction. mPGES-1, the enzyme responsible for the final step in the synthesis of PGE_2_ out of PGH_2_ [40], could not be investigated on protein levels in the current study, since SK-N-SH cells showed a high basal mPGES-1 signal (data not shown). This might be explained by the cross reaction of the available antibodies, potentially binding to cPGES or mPGES-2 as well and, therefore, pretending high basal mPGES-1 protein levels. Thus, the observed reduction of PGE_2_ in SK-N-SH cells is likely and at least in part mediated by decreased COX-2 protein levels. The role of mPGES-1 in the observed decrease of PGE_2_ after KIT C and KIT H pretreatment, however, needs to be evaluated in future studies. Since COX-1 expression was not significantly affected by KIT C or KIT H compared to the untreated control, gastrointestinal ulcers as a side effect of COX-1 inhibition in vivo may be prevented. However, COX-1 protein levels were not examined in the current studies. The observed effects of KIT C and KIT H in LPS-stimulated primary mice microglia are likely to be mediated by similar mechanisms as in IL-1β-stimulated SK-N-SH cells. Therefore, further studies are necessary to investigate the effects of coumarin derivates on COX-1 protein synthesis and gastrointestinal effects as well as the responsible pathways and mechanisms in primary microglia. The reduction of the pro-inflammatory IL-6 in LPS-stimulated primary microglia and the enhanced expression of the anti-inflammatory IL-10 in IL-1β-stimulated SK-N-SH cells after treatment with KIT C and KIT H supports the anti-inflammatory potential of these coumarin compounds. Since the results of KIT C and KIT H in SK-N-SH cells as well as the preliminary data in primary mice microglia are promising, future experiments with both compounds using animals are ethically justifiable.

The signal transduction induced by GPR55 activation and targeted by KIT C and H resulting in the shown COX-2 protein reduction might involve different downstream pathways associated with the GPR55. We analyzed different downstream pathways of the GPR55 using KIT 10 in primary microglia demonstrating no effects on the mitogen activated protein kinases (MAPK)-pathway but a reduced LPS-induced IκB-α phosphorylation [14]. As shown in previous publications, PGE_2_-suppression as well as decreasing COX-2 protein levels can be achieved by inhibiting NF-κB- and MAPK-pathways [41]. For the current study, different pathways, such as MAP-kinases (phosphor-Erk1/2, p38 MAPK, SAPK/Jnk), IκB-α, NF-κB, protein kinase C (PKC), nuclear factor of activated t-cells (NFAT), nuclear growth factor (NGF), and brain-derived neurotrophic factor (BDNF) were examined, but neither KIT C nor KIT H significantly affected the investigated pathways (data not shown).

The inhibition of oxidative stress/isoprostane pathways independent of COX-2 might be responsible for the reduction of PGE_2_ by KIT C and KIT H as well, as discussed in our previous study [42] showing that COX enzymatic activity is enhanced by antioxidative stress [43]. In this and a previous study, KIT C as well as KIT H decreased oxidative stress and 8-Iso-PGF_2α_ synthesis via GPR55 coupled to the inhibition of PGE_2_ and therefore possibly affecting the AA pathway by reduced COX-activity [17]. However, COX enzymatic activity is not significantly affected by treatment with KIT C or KIT H in the current study. Inhibition of phospholipase A2 (PLA2) reduces the concentration of available AA in the cells; therefore, PGE_2_-levels are reduced as a consequence [44]. In the current study, we did not assess PLA2 expression, protein levels, or activity, so we are not able to exclude a PLA2 dependent mechanism of the PGE_2_-suppression. For this reason, the molecular targets of KIT C and KIT H leading to PGE2/COX-2 reduction need to be further investigated in future studies.

In contrast to the decrease of COX-2 protein levels in 24-h IL-1β-stimulated SK-N-SH cells, mRNA concentrations were significantly increased due to the treatment with KIT C and as tendence after treatment with KIT H in 4-h stimulated cells. We conducted a time curve with 10 µM of both compounds, investigating protein as well as mRNA levels after 2, 4, 8, 12 and 24 h of IL-1β-stimulation. After 2 and 4 h of stimulation, we were not able to detect COX-2 by Western Blot. Protein levels of KIT C and KIT H pretreated cells were increased compared to IL-1β only treated cells at 8 (KIT C and KIT H) and 12 h (KIT C only). KIT H pretreated cells showed less COX-2 protein after 12 h of stimulation. COX-2 mRNA levels were significantly increased by both KITs after only 4 h of stimulation but were higher or at least comparable to the IL-1β positive control at all time points. Therefore, the difference between mRNA levels and protein levels of COX-2 cannot be explained by the dynamic over time. It has been shown that COX-2 mRNA stability is mediated by Erk1/2 and can be affected by G-protein coupled receptors (GPCRs). The Kaposi sarcoma virus oncogenic protein (vGPCR), for example, is associated with COX-2 overexpression and enhances mRNA stability of COX-2 [45]. Reduced mRNA-stability, however, leads to an earlier degradation of COX-2 mRNA, so even higher levels of mRNA might not usher high levels of COX-2 protein, because the mRNA is shorter and available for translation. Since the inhibition of pathways such as Erk1/2 coupled to the GPR55 has been shown to affect mRNA stability, this could be a possible explanation for increased COX-2 mRNA levels not leading to increased protein levels. However, phosphorylation of Erk1/2 is not affected by treatment with KIT C or KIT H in SK-N-SH-cells (not shown). Another mechanism leading to reduced COX-2 protein levels with increased mRNA concentrations might be a ubiquitin-mediated degradation of COX-2 protein. It has been shown that Centromere Protein U (CENPU) reduces ubiquitination as well as degradation of COX-2 in breast cancer [46]. Even if CENPU-dependent affection of COX-2 degradation is not likely to be responsible for the observed effects of KIT C and KIT H on COX-2 protein and mRNA levels, the mechanism of enhanced COX-2 degradation might be triggered by other pathways such as protein kinase C. Further studies are needed to elucidate the underlying mechanisms of the observed effects of KIT C and H on the AA pathway.

The observed COX-2 protein levels as well as PGE_2_ reduction after 24 h of IL-1β-stimulation is not accompanied by a reduction of COX-1 or COX-2 enzymatic activity. The enzymatic activity was measured after 30 min of treatment with KIT C and KIT H, so the enzymatic activity is independent of changes to protein levels of COX-1 or COX-2, as demonstrated in the time curve (Appendix A). The time curves demonstrate that changes of COX-2 levels can be found earliest after 8 h. The results show that KIT C and KIT H do not specifically bind and inhibit COX-1 or COX-2 enzymes, but rather affect the enzyme synthesis via pathways coupled to the GPR55.

It has been shown and suggested that coumarin derivates (KITs) enfold their biological effects by GPR55 antagonisms with inverse agonistic activity at the GPR55 [13,14,16]. However, neither for KIT C nor for KIT H radioligand assays have been implemented. Instead, a GPR55 activation assay was performed demonstrating a biological activity of KIT C and KIT H at the receptor, comparable to the known agonists LPI and AM251. The GPR55 activation data support the hypothesis that KIT C and KIT H enfold their effects by acting via direct activation of GPR55. Therefore, we further studied the effects of the commercially available GPR55 agonist O-1602 as well as the GPR55 antagonist ML 193. Interestingly, the GPR55 antagonist ML 193, but not the agonist O-1602, showed comparable biological effects to KIT C and KIT H in most experiments, suggesting an antagonistic activity of ML 193 with additional inverse agonistic activity like KIT C and KIT H at the GPR55 as shown with the previously studied coumarin derivates [16,21]. In line with this hypothesis, the decrease of IL-1β-induced PGE_2_-levels after ML 193 treatment remains between the effect size of PGE_2_-inhibition due to KIT C and KIT H administration. Our previous study, investigating the anti-oxidative effects of KIT C and KIT H in SK-N-SH cells, furthermore demonstrated the GPR55-dependency of the effects by both compounds since the antioxidative effects of KIT C and KIT H were abolished after GPR55 knockout in SK-N-SH cells [17]. The molecular structure of ligands determines either agonistic or antagonistic (respective with inverse agonistic activity) effects at the receptor [19]. All KITs share the coumarin scaffold with different chemical residues, which might momentously change the biological effects via GPR55. This might be an explanation of the observed differences between the effect size of KIT C and KIT H in different experiments. In a previous study, the efficacy of KIT 3, KIT 17, and KIT 21 showed enormous differences in the magnitude of inhibition of LPS-induced PGE_2_-release [13], underlining the hypothesized importance of the chemical residues. The chemical residues and the distribution of electronegativity in the molecules determine the position of the molecule and the deepness of binding in the GPR55 binding pocket and, therefore, the triggered effects [19]. Furthermore, heterocyclic or aromatic residues are described as a common characteristic of GPR55 antagonists, protruding out of the binding pocket and stabilizing the receptors “Off”-conformation [19], which at least KIT H exhibits. Since we observe effects that suggest an inverse agonistic activity of the substances, the lack of the heterocyclic or aromatic ring of KIT C basically does not question its GPR55-affinity as antagonist with inverse agonistic activity. On a biomolecular level, functional selectivity and differences in conformational change [37,38] after the binding of the compounds to GPR55 due to the differences in the chemical residues might be the explanation for the different effects of the tested coumarin derivates. However, the underlying mechanisms of the observed differences of the effects of KIT C and KIT H cannot be answered by the recent study. Therefore, future research is necessary to identify the exact role of the chemical residues in the biological effectiveness. This might open new options in the pharmacological drug design based on the coumarin scaffold.

Interestingly, the effects of KIT C and KIT H vary dependent on the cell type used as well. For example, KIT C but not KIT H showed a significant reduction of IL-1β-induced PGE_2_ in primary mice microglia. In SK-N-SH cells, however, KIT H showed a stronger inhibition of PGE_2_ than KIT C. Since we compared two different types of cells, namely neuroblastoma (neuronal) and microglial cells, the observed digressing effects might be explained by differences in GPR55 density on the cell surface, coupled Gα proteins, and associated downstream pathways as shown for other GPCRs. For example, for muscarinic agonists such as carbachol, weak efficacy in combination with differing levels of receptor affinity leads to cell- or tissue-dependent selectivity at the muscarinic GPCRs [37]. Furthermore, the proposed differences in receptor density and G-protein coupling dependent on the cell cycle phase has been shown for calcitonin before [47]. Since the investigated cells take over different functions in the CNS, with microglia fulfilling a key function in immune defense and neuroinflammation [48], the different effects might be explained by the different physiological cell functions. Neurons are primarily affected by the microglial inflammatory response [48] and physiologically building neuronal networks for signal transduction and processing [49].

Another possible explanation for the observed cell-dependent effects of KIT C and KIT H might be the cells’ donor species. In this study, we compared human neuroblastoma cells with primary microglia cells of C57B1/6 WT mice. The cloning of the GPR55 showed 4.3, 7, and 10 kilobase (kb) mRNA transcripts in human brain tissue whereas the examination of the rat spleen revealed a 3.5 kb mRNA transcript [18]. A search in the NCBI Protein Database comparing GPR55-sequences evinced small differences in the number of amino acids. In humans, 319 amino acids form the GPR55, and in mice, the receptor is built by 327 amino acids. Therefore, the receptor most likely reveals differences between humans and mice, and for this reason, the same compound might elicit different effects depending on the species investigated. Species-selective effects in the endocannabinoid system have been shown for rats and mice investigating the cannabinoid receptor (CB)2. Rats and mice were treated with intranasal JWH-133, a selective CB2 agonist, and mice but not rats showed a reduction in the self-administration of intravenous cocaine as a consequence [50]. A third possible mechanism explaining the observed species-differences might be the expression and protein synthesis of the GPR55 as well as the coupled Gα proteins and pathways that may vary between species, as mentioned for cell-type selective effects earlier [37]. Further studies are necessary to distinguish the proposed mechanisms of the cell-/species-selective effects of KIT C and KIT H. Since studies in model organisms are often required before clinical studies can be implemented, differences between species are important to draw the right conclusions about doses and expected effects in humans.

The used concentrations of KIT C and KIT H as well as O-1602 and ML 193 were chosen based on previous experiences with similar coumarin derivates [13,14,17] as well as publications investigating the commercial GPR55 agonist and antagonist [51,52,53,54]. O-1602 is a potent agonist at the GPR55 with an EC_50_ of 13 nm [55], and ML 193 is a potent antagonist of the GPR55 with a GPR55-potency of 221 nm [29]. IC_50_ of other related coumarin-derivates such as KIT 10 [14] and KIT 17 [13] was between approximately 5 µM and 10 µM in radioligand-assays. The IC_50_ data of KIT 10 and Kit 17 were determined using a cellular system and was therefore in the µM range when compared to ML 193 that was determined using membrane preparation. Unfortunately, for KIT C and KIT H, no IC_50_ values are available, and for the GPR55 and the investigated compounds no B_max_ or K_D_ values are available. However, the used concentrations of O-1602 and ML-193 are much higher than the reported EC_50_ and IC_50_ values. Therefore, the observed effects for at least those two compounds might not only be GPR55-dependent but mediated by specific intermolecular actions. Specific interactions are defined as ligand-receptor-binding, whereas non-specific interactions can occur between ligands and other enzymes or proteins of the cells that are not the targeted receptor [56,57]. The used concentrations of KIT C and KIT H, in contrast, are at sufficient levels compared to the IC_50_ values of other coumarin based compounds. Moreover, for KIT C and KIT H, knockout of the GPR55 in SK-N-SH-cells, as described in a previously published paper, fully abolished observed effects of both coumarin compounds on oxidative stress-induced cell death [17]. This, and the data obtained from our GPR55 activity assay, support the hypothesis that effects of KIT C and KIT H are GPR55 dependent and not non-specific receptor-independent interactions, even if they have lower affinity to the GPR55 than the commercially available ligands used.

A growing body of research focuses on anti-inflammatory treatment in numerous neurological and psychiatric diseases. For depression, positive effects of NSAIDs and glucocorticoids were observed [5], but NSAIDs are associated with gastrointestinal ulcers, and long glucocorticoid treatments may cause numerous side effects, such as osteoporosis and an increase of infections [58]. Coumarin derivates with anti-inflammatory characteristics and possibly milder side effects might be an alternative to the already known anti-inflammatory drugs. Some authors support the hypothesis of the inflammatory genesis of depression by the fact that autoimmune diseases, as well as depression, are observed at about two times higher prevalence in women than in men [2]. Besides depression, anti-inflammatory drugs such as COX-2 inhibitors ameliorated psychiatric symptoms in schizophrenia, Borderline Personality Disorder, and obsessive-compulsive disorders, whereas NSAID treatment reduced the anti-depressive effects of selective serotonin reuptake inhibitors (SSRI) [4]. Furthermore, neuroinflammation is associated with neurological diseases, such as PD and AD, as well as neuroinflammation possibly disturbing the balanced interaction of different CNS cells and leading to neurodegeneration [3].

Anti-neuroinflammatory capacities have been shown for coumarin derivates in previous studies [13,14] as well as for KIT C and KIT H in the current study. Therefore, coumarin derivates such as KIT C as well as KIT H might be promising novel compounds in the treatment of diseases with (neuro)inflammatory etiologies and pathomechanisms. Further understanding of functional selectivity [37,38] at the GPR55 based on different chemical residues as well as species differences might improve pharmaceutical drug design for these diseases. The evaluation of KIT C and KIT H in animal disease models might improve the understanding of the potential of both compounds in the future treatment of neurological and psychiatric diseases and demonstrate possible side-effects of coumarin derivates.

## 4. Materials and Methods

### 4.1. Chemicals

KIT C and KIT H were synthesized as described previously [17] by the Karlsruher Institute for Technology (KIT), Institute of Organic Chemistry, dissolved in DMSO (Merck KGaA, Darmstadt, Germany) and used in final concentrations of 0.1–25 µM. The GPR55 agonist O-1602 and GPR55 antagonist ML 193 (both from Cayman Chemicals, Ann Arbor, MI, USA, distributed by BioMol, Hamburg, Germany) were dissolved in DMSO and used in final concentrations of 5 µM (O-1602) and 25 µM (ML 193). Human interleukin (IL)-1β (100,000 U/mL in phosphate buffered saline (PBS)) was purchased from Roche Diagnostics (Manheim, Germany) and was used at a final concentration of 10 U/mL for the experiments. 5 mg/mL lipopolysaccharide (LPS) from Salmonella typhimurium (Sigma-Aldrich GmbH, Taufkirchen, Germany) was dissolved in PBS as stock and diluted with distilled water for a final concentration of 10 ng/mL in primary microglia cultures.

Figure 9 shows the chemical structure of KIT C and KIT H that were already introduced in a previous paper [17], as well as the structures of O-1602 and ML 193 obtained from the manufacturer’s (Cayman Chemicals, Ann Arbor, MI, USA) website.

### 4.2. Human Neuroblastoma (SK-N-SH) Cell Culture

SK-N-SH neuronal cells were obtained from the American Type Culture Collection (HTB-11, Rockville, MD, USA) and approximately 2 × 10^6^ cells were grown in 1× minimum essential medium (MEM) containing Earl’s salts, 10% fetal bovine serum (Bio & SELL GmbH, Feucht/Nürnberg, Germany), 1 mM l-glutamine, 1 mM sodium pyruvate, 2 mL of 100× MEM vitamin solution, 40 units/mL penicillin, 40 µg/mL streptomycin and 0.1 µg/mL fungizone^®^ (all obtained from Gibco, Thermo Fisher Scientific, Bonn, Germany). Cells were incubated at 37 °C in a humidified atmosphere with 5% CO_2_ in 75 cm^2^ cell culture flasks (Falcon, Heidelberg, Germany) with approximately 2 × 10^5^ cells per cm^2^ when splitting the sub-cultures and grown to about 90% confluency. Confluent monolayers were passaged routinely by trypsinization. After trypsinization, cells were harvested and re-seeded into 6-, 12-, 24- or 96-well plates (between 2.8 × 10^5^ and 2 × 10^4^ cells depending on the plate size). On the next day, the medium was changed after 1 h, and cells were stimulated for respective experiments.

### 4.3. Primary Mouse Microglial Cell Cultures

#### 4.3.1. Ethics Statement

Animals were obtained from the Center for Experimental Models and Transgenic Services-Freiburg (CEMT-FR). All the experiments were approved and conducted according to the guidelines of the ethics committee of the University of Freiburg Medical School under protocol No. X-19/06R and the study was carefully planned to minimize the number of animals used and their suffering.

#### 4.3.2. Primary Mouse Microglia Cultures

Primary mouse mixed glia cultures were prepared from 2 to 3 days old C57B1/6 WT mice as described before [44,59,60,61]. Briefly, brains were carefully taken under sterile conditions and meninges were removed. The cortices were dissociated and filtered through a 70 µm nylon cell strainer (BD biosciences, Heidelberg, Germany). After centrifugation at 1000 rpm for 10 min the cells were resuspended in LPS-free Dulbecco’s modified Eagle’s medium (DMEM) with 10% fetal calf serum (FCS; Bio & SELL GmbH, Nürnberg/Feucht, Germany) and antibiotics (DMEM and anti-anti obtained from Gibco, Thermo Fisher Scientific, Bonn, Germany) and cultured in 10 cm cell culture dishes at a density of 5 × 10^5^ cells/plate (Falcon, Heidelberg, Germany) in the humified atmosphere at 10% CO_2_ and 37 °C. After 12 days in vitro, floating microglia were harvested by gently shaking on an orbital shaker and re-seeded into 6-well plates with approximately 3 × 10^5^ cells per well and an 80% survival rate to obtain pure microglial cell cultures. The protocol for preparing pure primary microglial cultures was established by Gebicke-Härter et al. in our laboratory and published in 1989. This publication demonstrated a purity of >98% for microglia using morphological features, immunofluorescence (monocyte/macrophage marker anti-CD68 = ED1), and cytochemical analysis [60]. On the next day, the medium was changed to remove non-adherent cells, and after 1 h, cells were stimulated for the experiments.

### 4.4. Cell Viability Assay

MTT assay (Sigma-Aldrich GmbH, Taufkirchen, Germany) was used for measuring the viability of SK-N-SH neuronal cells after treatment with KIT C (1, 5 and 10 µM), KIT H (1, 5 and 10 µM), O-1602 (5µM), or ML 193 (25 µM). This assay determines the number of metabolically active cells and allows conclusions about viable cells in the culture based on the reduction of a yellow tetrazolium salt (3-(4,5-dimethylthiazol-2-yl)-2,5-diphenyltetrazolium bromide or MTT) to purple formazan in the cells. Briefly, cells were cultured in 96-well plates at a density of 25 × 10^3^ cells/well for 24 h. The medium was changed and after at least 1 h, cells were pre-treated with different concentrations of the compounds for 30 min. Cells were then incubated with or without IL-1β for the next 20 h. 20 µL Ethanol (approximately 20% end conc.) was used to induce cell death as a positive control. Next, 20 µL of MTT-solution (working concentration 5 mg/mL) were added to all wells and incubated for another 4 h at 37 °C. Afterwards, the medium was removed and replaced with 200 µL of DMSO. The colorimetric reaction was measured using the MRX^e^ Microplate reader (Dynex Technologies, Denkerdorf, Germany) at 595 nm.

### 4.5. Determination of PGE_2_-Release

SK-N-SH neuronal cells were pretreated with KIT C (0.1–25 µM), KIT H (0.1–25 µM), O-1602 (5 µM), or ML 193 (25 µM) for 30 min. Afterwards, cells were incubated for the next 24 h with or without IL-1β (10 U/mL) and supernatants were collected. The levels of PGE_2_ were measured using a commercially available enzyme immunoassay (EIA) kit (from Cayman Chemicals, Ann Arbor, MI, USA, distributed by BioMol, Hamburg, Germany) following the manufacturer’s protocol. The results were normalized to IL-1β and presented as a percentage of change in PGE_2_ levels of at least three independent experiments.

PGE_2_-release in primary microglia cultures was measured as described for SK-N-SH cells. Only KIT C (5 and 10 µM) and KIT H (5 and 10 µM) were evaluated in primary microglia and LPS (10 ng/mL) was used for stimulation instead of IL-1β. Supernatants were collected after 24 h and used in the EIA following the protocol. The results were normalized to LPS and presented as percentage of change in prostaglandin levels of at least three independent experiments.

### 4.6. Determination of GPR55 Agonistic Activity

The determination of GPR55 agonistic activity was carried out using HEK293T-GPR55 cells that overexpress the human GPR55 [62]. Briefly, HEK293T-GPR55 cells were cultured in 24-well plates (10^5^ cells/well) and transiently transfected with 0.2 µg of the reporter plasmid CRE-Luc that contains six consensus cAMP responsive elements (CRE) linked to firefly luciferase reporter gene using Roti©-Fect (Carl Roth, Karlsruhe, Germany). Transfected cells were treated with increasing concentrations of either the test compounds KIT C and KIT H or the positive controls AM251 and LPI for 6 h. Then, the cells were washed twice with PBS 1× and lysed in 100 µL lysis buffer containing 25 mM Tris-phosphate (pH 7.8), 8 mM MgCl_2_, 1 mM DTT, 1% Triton X-100, and 7% glycerol for 15 min at room temperature in a horizontal shaker. Luciferase activity was measured using a TriStar2 Berthold/LB942 multimode reader (Berthold Technologies, Bad Wildbad, Germany) following the instructions of the luciferase assay kit (Promega, Madison, WI, USA). The relative light units (RLUs) were calculated, and the results were expressed as the percentage of activation over the control. The experiment was performed three times.

### 4.7. RNA Isolation and Quantitative PCR

For quantification of the mRNA of the enzymes of the COX-2/PGE_2_ pathway, we performed quantitative real-time PCR (qPCR) in SK-N-SH cells. Cultured cells were pretreated with KIT C or KIT H (0.1–10 µM) for 30 min, followed by stimulation with IL-1β (10 U/mL) for 4 h. For the COX-2 time curve, cells were incubated with IL-1β for 2, 8, 12 and 24 h, respectively. Total RNA was extracted using the GeneMATRIX Universal RNA Purification Kit (Roboklon GmbH, Berlin, Deutschland) according to the manufacturer’s protocol. Then, cDNA was reverse transcribed from 500 ng of total RNA in a 30 μL total reaction volume with an initial denaturation at 70 °C (10 min) with the following amplification cycle after the addition of the master mix. The qPCR amplification was carried out by the CFX96 real-time PCR detection system (Bio-Rad Laboratories GmbH, Feldkirchen, Germany). Glyceraldehyde 3-phosphate dehydrogenase (GAPDH) served as an internal control for sample normalization. The primer sequences were GAPDH: Forward (Fwd): 5′-TGGGAAGCTGGTCATCAAC-3′/Reverse (Rev): 5′-GCATCACCCCATTTGATGTT-3′, COX-2: Fwd: 5′-CTTCACGCATTTCAAG-3′/Rev: 5′-TCACCGTAAAGTCCAC-3′, mPGES-1: Fwd 5′-TGCAGCACGCTGCTGGTCAT-3′/Rev 5′-GTCGTTGCGGTGGGCTCTGAG-3′, COX-1: Fw: 5′-TCCATGTTGGCTATGG-3′/Rv: 5′-GTGGTGGTCCTTCCTG-3′ and IL-10: Fwd: 5′-GATGCCTTCAAGTGAA-3′/Rev: 5′-GCAACCCAGGCTTAAA-3′. Primers were designed using Universal ProbeLibrary Assay Design Center (Roche Diagnostics, Mannheim, Germany) and obtained from biomers.net GmbH (Ulm, Germany).

### 4.8. Immunoblotting

SK-N-SH neuronal cells were pre-treated with KIT C and KIT H (0.1–10 µM) for 30 min. After 24 h (COX-2) of IL-1β-stimulation (10 U/mL), cells were washed with cold PBS and lysed mechanically in lysis buffer (42 mM Tris–HCl, 1.3% sodium dodecyl sulfate, 6.5% glycerin, 100 μM sodium orthovanadate, and 2% phosphatase and 0.2% protease inhibitors). For the COX-2 time curve, cells were incubated with IL-1β for 2, 4, 8 and 12 h, respectively. Protein concentrations of the samples were measured using the bicinchoninic acid (BCA) protein assay kit (Thermo Fisher Scientific, Bonn, Germany). For Western blotting, 20 μg of total protein from each sample were subjected to sodium dodecyl sulfate-polyacrylamide gel electrophoresis (SDS-PAGE) under reducing conditions. Proteins were then transferred onto polyvinylidene fluoride (PVDF) membranes (Merck Millipore, Darmstadt, Germany) by semi-dry blotting. After blocking with Roti-Block (Roth, Karlsruhe, Germany), membranes were incubated overnight with primary antibodies (mouse anti-COX-2 (MAB-4198, 1:1000; R&D systems, Wiesbaden, Germany), mouse anti-actin (sc-47778, 1:5000, Santa Cruz Biotechnology Inc., Heidelberg, Germany)). The proteins were detected with horseradish peroxidase-coupled sheep anti-mouse IgG (1:20,000 dilution; Amersham Biosciences GmbH, Freiburg, Germany) using enhanced chemiluminescence (ECL) reagents (Biozym, Hessisch Oldendorf, Germany) and taking pictures with the ChemiDoc MP imaging system (Bio-Rad Laboratories GmbH, Feldkirchen, Germany). Densitometric analysis was performed using ImageJ software (V1.48t, NIH, Bethesda, MD, USA).

### 4.9. Cyclooxygenase Activity Assay

COX enzymatic activity was investigated using the arachidonic acid assay, as described previously [63]. For COX-1 activity, neuroblastoma cells were plated in 24-well plates and after 24 h, the medium was removed and replaced with a serum-free medium. KIT C or KIT H (0.1–10 µM) or the selective inhibitor of COX-1 SC560 [(1 and 10 μM); Sigma-Aldrich GmbH, Taufkirchen, Germany] was added and left for 15 min. Then, arachidonic acid (15 µM; Sigma-Aldrich GmbH, Taufkirchen, Germany) was applied for another 15 min. Finally, supernatants were collected and used for the determination of PGE_2_ as described above.

For COX-2 enzymatic activity, the assay was conducted as for COX-1, but with pre-incubation of IL-1β (10 U/mL) for 24 h to induce COX-2 synthesis and using diclofenac sodium ((0.1 and 1 μM); Sigma-Aldrich GmbH, Taufkirchen, Germany) as commercial COX-2 preferential inhibitor.

### 4.10. Determination of IL-6 Release

Effects of KIT C and KIT H (0.1–25 µM) on IL-6 release in IL-1β-stimulated SK-N-SH cells and (concentrations 5 and 10 µM) in LPS-stimulated primary microglia were evaluated using ELISA. Commercially available Invitrogen^TM^ eBioscience^TM^ human or mouse IL-6 ELISA Ready-SET-Go!^TM^ Kits (Thermo Fisher Scientific, Bonn, Germany) were used following the manufacturer’s protocol. Briefly, cells were stimulated after pretreatment with KIT C or KIT H for 24 h with IL-1β (SK-N-SH cells) or LPS (primary microglial cells). Supernatants were collected and stored at −80 °C for further experiments. ELISA-plates (Nunc MaxiSorp^TM^; Thermo Fisher Scientific, Bonn, Germany) were coated with IL-6 capture antibody overnight. The next day, samples were added followed by the addition of an IL-6 determination antibody after removing supernatants and washing the plate. The amount of bound IL-6 determination antibody was quantified using an HRP-dependent colorimetric reaction. The absorbance of the wells was read at 450 nm using the MRX^e^ Microplate reader and calculated as % of IL-1β or LPS after blank substraction.

### 4.11. Statistical Analysis

Raw values were converted to percentage and IL-1β (10 U/mL), LPS (10 ng/mL), or the appropriate negative control, such as untreated cells for MTT-assay, were considered as 100%. Data are represented as mean ± SEM of at least three independent experiments. The statistical comparisons were performed using one-way ANOVA with Dunett’s post hoc test (Prism 8 software, GraphPad software Inc., San Diego, CA, USA). The level of significance was set at * *p* < 0.05, ** *p* < 0.01, *** *p* < 0.001 and **** *p* < 0.0001 and is indicated in the figures.

## 5. Conclusions

Anti-inflammatory treatment of neurological and psychiatric diseases has been shown to abate disease’s symptoms in murine models and clinical studies. However, many available anti-inflammatory drugs are associated with momentous side effects. Therefore, the development of new pharmacological therapeutics might improve the treatment of these CNS diseases. KIT C and KIT H, likely enfolding their effects via GPR55, might be promising novel anti-inflammatory strategies for future interventions by decreasing pro-inflammatory PGE_2_, IL-6, and increasing anti-inflammatory IL-10. Future research is necessary to fully understand the involved mechanisms and effects of KIT C and KIT H in in vivo models and following clinical studies.

## Figures and Tables

**Figure 1 ijms-23-00959-f001:**
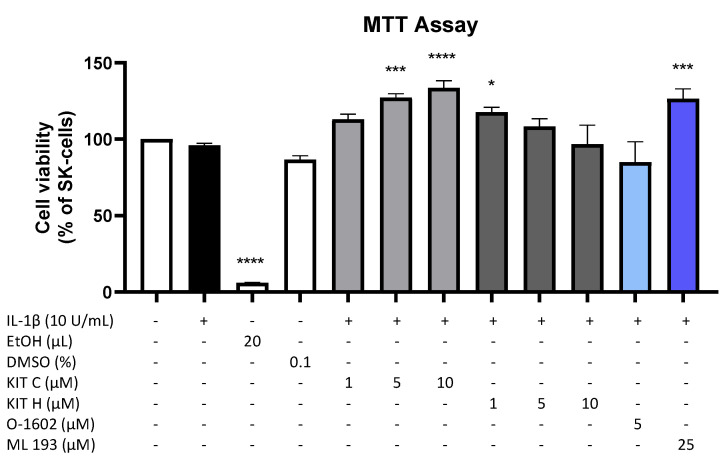
Effects of KIT C (light grey bars), KIT H (dark grey bars), O-1602 (light blue bar), and ML 193 (blue bar) on cell viability in IL-1β-stimulated SK-N-SH cells (24 h treatment). Cell viability was measured by change in color due to MTT-oxidation, and absorbance was measured at 595 nm using an ELISA-reader. Values are presented as the mean ± SEM of at least three independent experiments. Statistical analysis was performed using one-way ANOVA with Dunett’s post hoc tests with * *p* < 0.05, *** *p* < 0.001, **** *p* < 0.0001 compared to untreated cells. The figure is derived from our previous publication [17].

**Figure 2 ijms-23-00959-f002:**
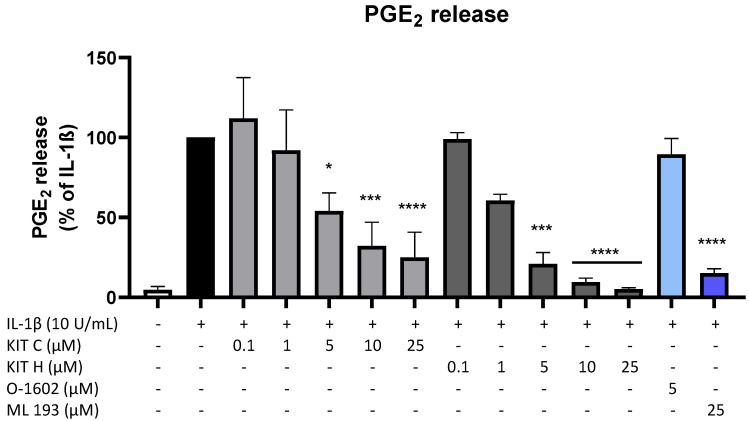
Effects of KIT C (light grey bars), KIT H (dark grey bars), O-1602 (light blue bar), and ML 193 (blue bar) on PGE_2_-release in IL-1β-stimulated SK-N-SH cells. Cells were stimulated as described under material and methods. After 24 h of stimulation, supernatants were collected and the release of PGE_2_ was measured by EIA. Values are presented as the mean ± SEM of at least three independent experiments. Statistical analysis was performed using one-way ANOVA with Dunnett’s post hoc tests with * *p* < 0.05, *** *p* < 0.001, **** *p* < 0.0001 compared to IL-1β.

**Figure 3 ijms-23-00959-f003:**
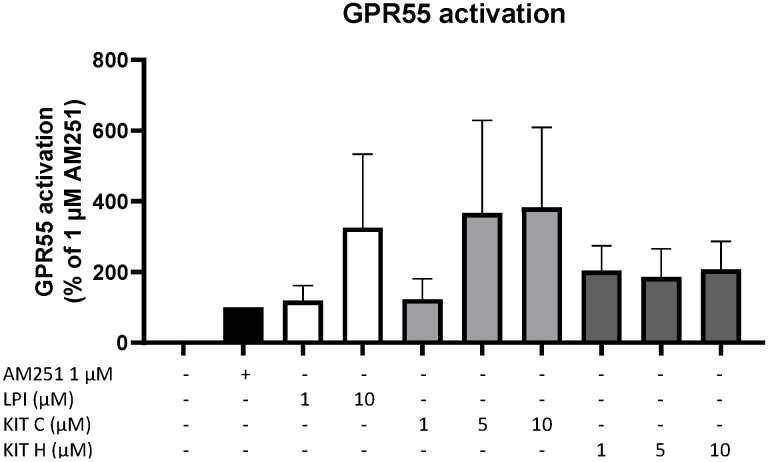
GPR55 activation by AM251 (black bar), LPI (white bars), KIT C (light grey bars), and KIT H (dark grey bars) in HEK293T-GPR55 cells. Cells were treated as described under material and methods. After 6 h of stimulation, cells were lysed, and the luciferase activity was measured. Values are presented as the mean ± SEM of at least three independent experiments. Statistical analysis was performed using one-way ANOVA with Dunnett’s post hoc tests compared to 1 µM AM251.

**Figure 4 ijms-23-00959-f004:**
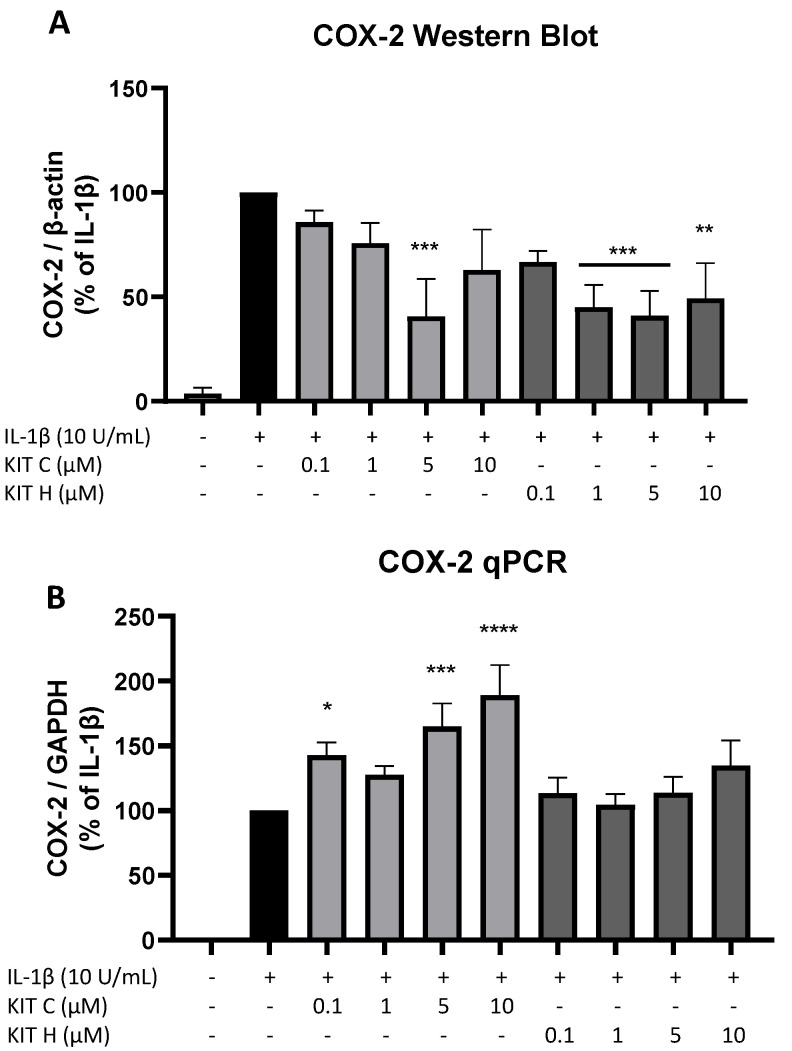
Effects of KIT C (light grey bars) and KIT H (dark grey bars) on COX-2 protein levels (**A**) and COX-2 mRNA expression (**B**) in IL-1β-stimulated SK-N-SH cells. Cells were stimulated and analyzed as described under material and methods. Values are presented as the mean ± SEM of at least three independent experiments. Statistical analysis was performed using one-way ANOVA with Dunnett’s post hoc tests with * *p* < 0.05, ** *p* < 0.01, *** *p* < 0.001 and **** *p* < 0.0001 compared to IL-1β.

**Figure 5 ijms-23-00959-f005:**
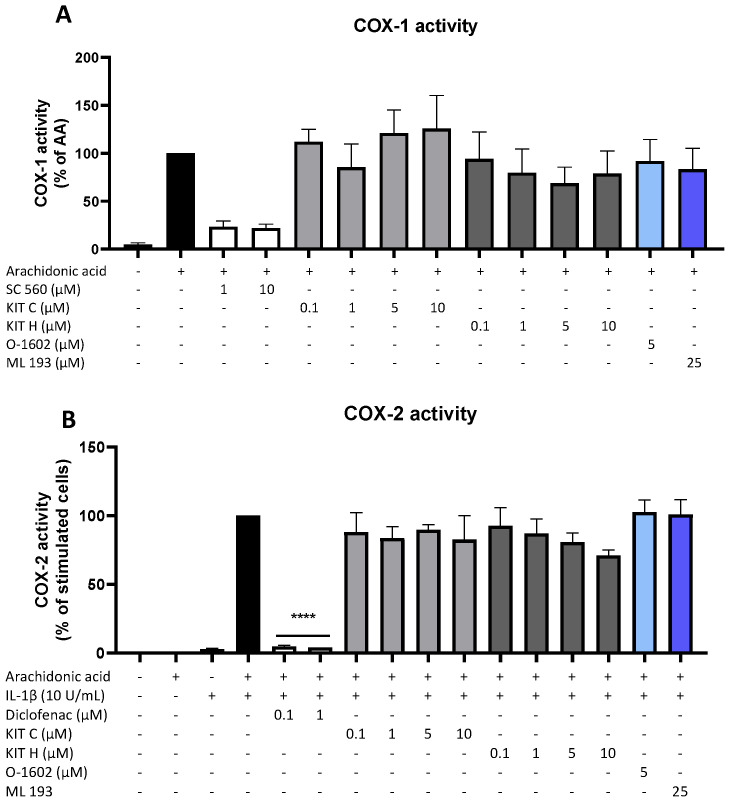
Effects of KIT C (light grey bars), KIT H (dark grey bars), O-1602 (light blue bar), and ML 193 (blue bar) on COX-1 (**A**) and COX-2 enzyme activities (**B**) in SK-N-SH cells. (**A**) COX-1-activity was measured after 15 min of incubation with arachidonic acid (AA). Levels of PGE_2_ in the supernatants were quantified by EIA. (**B**) After 24 h of IL-1β pre-stimulation, 15 µM of arachidonic acid (AA) was added and PGE_2_-release was measured by EIA. Values are presented as the mean ± SEM of at least three independent experiments. Statistical analysis was carried out by using one-way ANOVA with Dunnett’s post hoc tests with **** *p* < 0.0001 compared to AA (A) or IL-1β with AA (**B**).

**Figure 6 ijms-23-00959-f006:**
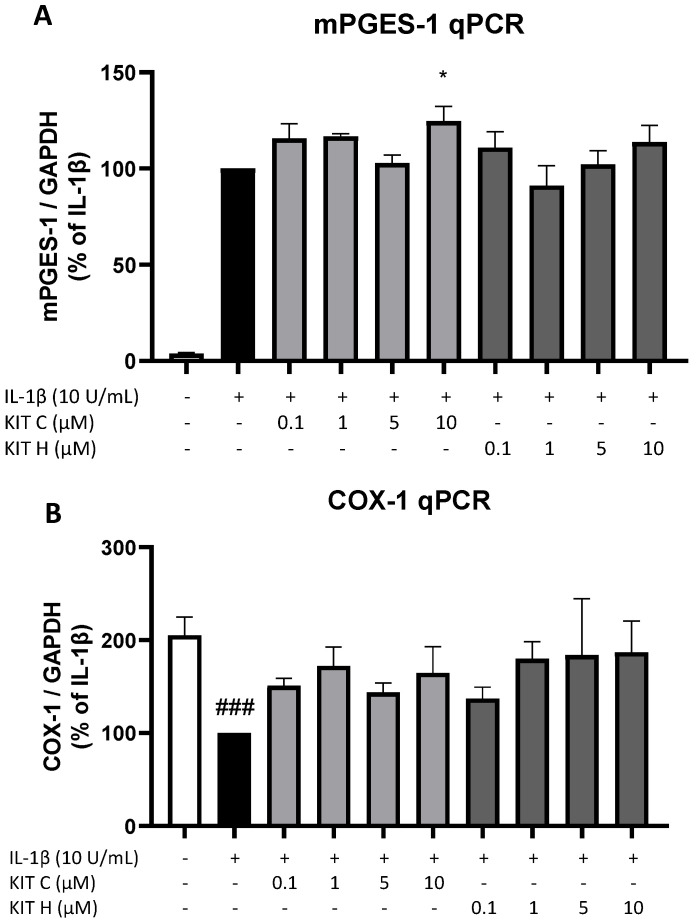
Effects of KIT C (light grey bars) and KIT H (dark grey bars) on mPGES-1 (**A**) and COX-1 mRNA expression (**B**) in IL-1β-stimulated SK-N-SH cells. Cells were stimulated as described under material and methods. After 4 h of stimulation, RNA was isolated and mRNA levels of the shown target genes were measured using qPCR. Values are presented as the mean ± SEM of at least three independent experiments. Statistical analysis was performed using one-way ANOVA with Dunnett’s post hoc tests with * *p* < 0.05 and ### *p* < 0.001 compared to IL-1β (**A**) or to untreated cells (**B**).

**Figure 7 ijms-23-00959-f007:**
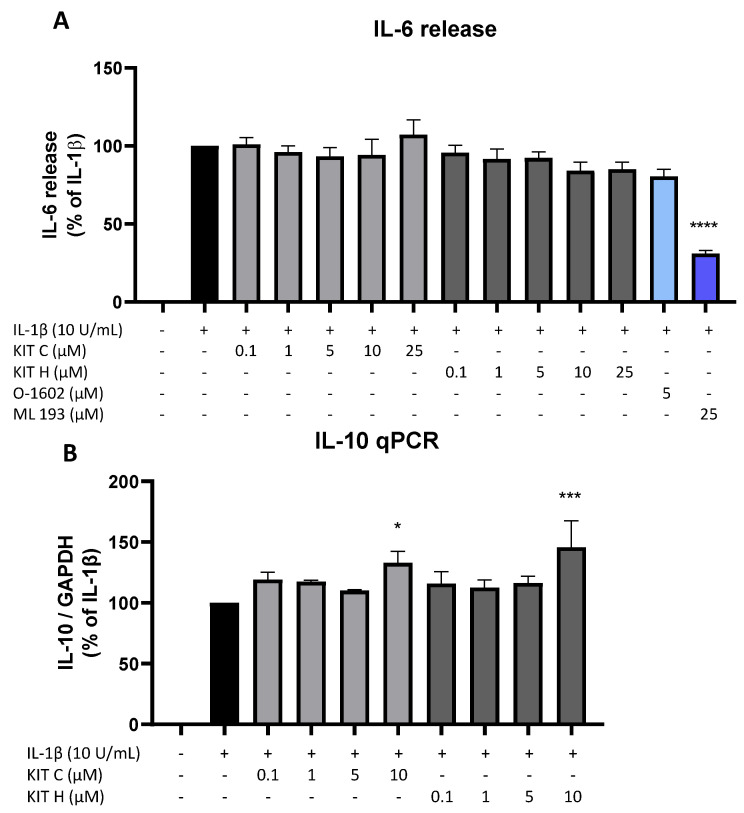
Effects of KIT C (light grey bars), KIT H (dark grey bars), O-1602 (light blue bar), and ML 193 (blue bar) on IL-6 release (**A**) and effects of KIT C and KIT H on IL-10 mRNA expression (**B**) in IL-1β-stimulated SK-N-SH cells. Cells were stimulated as described under material and methods. After 24 h of stimulation, supernatants were collected and the release of IL-6 was measured by ELISA (**A**). After 4 h of stimulation, RNA was isolated and the mRNA levels of the shown target genes were measured using qPCR. Values are presented as the mean ± SEM of at least three independent experiments. Statistical analysis was performed using one-way ANOVA with Dunnett’s post hoc tests with * *p* < 0.05, *** *p* < 0.001 and **** *p* < 0.0001 compared to IL-1β.

**Figure 8 ijms-23-00959-f008:**
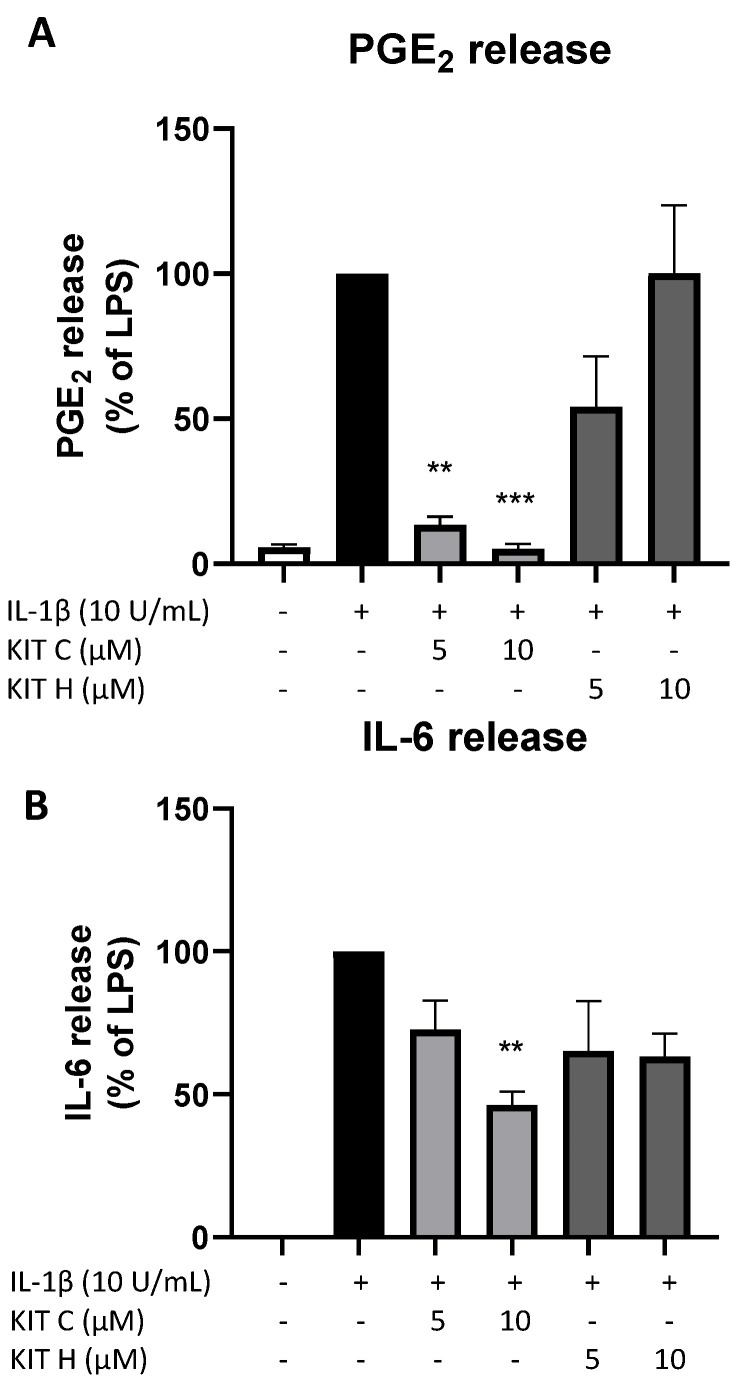
Effects of KIT C (light grey bars) and KIT H (dark grey bars) on PGE_2_- (**A**) and IL-6-synthesis (**B**) in LPS-stimulated primary mice microglia. Cells were stimulated as described under material and methods. After 24 h of stimulation, supernatants were collected and the release of PGE_2_ or IL-6 was measured by EIA/ELISA. Values are presented as the mean ± SEM of at least three independent experiments. Statistical analysis was performed using one-way ANOVA with Dunnett’s post hoc tests with ** *p* < 0.01 and *** *p* < 0.001 compared to LPS.

**Figure 9 ijms-23-00959-f009:**
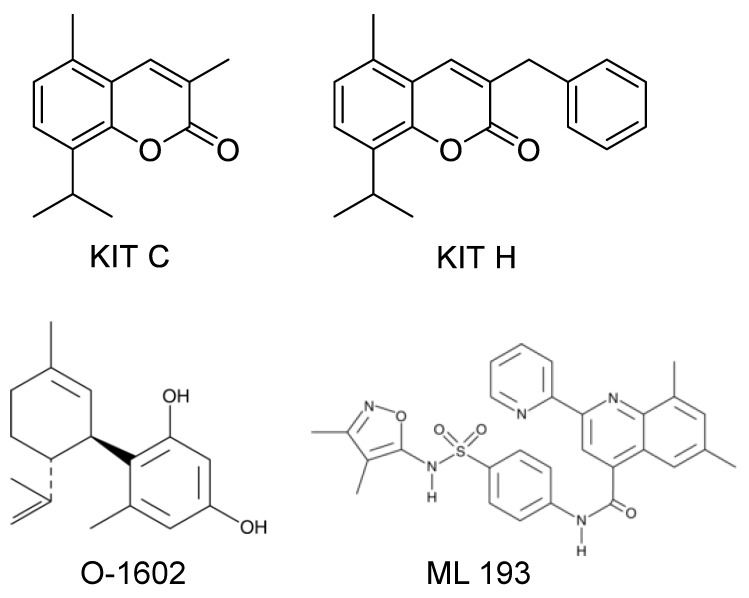
Chemical structures of KIT C, KIT H, O-1602, and ML 193 as previously published [17], respectively, as provided by the Cayman Chemicals website (https://www.caymanchem.com (accessed on 6 January 2022)).

## Data Availability

The data presented in this manuscript are available from the corresponding author upon request. The compounds used in this study were described before [17] and the data were submitted to the repository chemotion under the submission numbers CRS-9508 (KIT C) and CRS-9514 (KIT H). For a direct link to the data please refer to: KIT C (8-Isopropyl-3,5-dimethyl-2H-chromen-2-one): https://dx.doi.org/10.14272/LSYXPDGXGOSFCW-UHFFFAOYSA-N.1 (accessed on 14 January 2022) and KIT H (3-Benzyl-8-isopropyl-5-methyl-2H-chromen-2-one): https://dx.doi.org/10.14272/WNXYNMKVCJSJKO-UHFFFAOYSA-N.1 (accessed on 14 January 2022).

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
