# Peer review of "Functional Selectivity of Coumarin Derivates Acting via GPR55 in Neuroinflammation"

_ijms, 2022, doi:10.3390/ijms23020959_

Round 1

Reviewer 1 Report

Dear authors,

I have read the manuscript ijms-1548166 thoroughly. The submitted manuscript presents the results of an in-depth study on new coumarin derivatives and their mechanism of action in neuroinflammation. Overall, I believe the quality of the text is more than sufficient for the paper to be published in a high-ranking magazine, such as Int J Mol Sci, and may be of reasonably high interest for the readers. I have to say that the level of English used throughout the manuscript is superb and enables the reader to follow the story without additional problem caused by so-often poor presentation.

Abstract gives all the necessary information about the contents of the paper, keywords are appropriately chosen and all the necessary literature is listed as required. I have noticed that approximately 10 references out of 54 are self-citations, but in my opinion, this is not a problem, since they correctly refer to your previous work in this field.

Introduction gives all the necessary information for the readers. It may be a bit longer and too detailed in some places, but I do not insist on its shortening. There is only one detail that I believe would make the presentation of the whole paper even more “reader friendly”. As a medicinal chemist I am very fond of structures and the mere description of KITs and other compounds in Lines 102–159 gives little or almost no information on their structure. For old compounds, the reader may browse through the previous papers, but the process ids time consuming. Since KIT C and KIT H are novel molecules, I suppose their structures have not been disclosed yet. Figure with their structures is mandatory, I would suggest that chemical structures of KIT 10, KIT 17, O-1602 and ML 193 are also included.

Materials and methods section gives sufficiently data on research work and is well presented. The methodology of the research is sound. Results are clearly presented and supported with large number of Figures. The same is true with Discussion. All parts are written in such a detail, that I have no further suggestions.

Kind regards,

Author Response

Dear authors,

I have read the manuscript ijms-1548166 thoroughly. The submitted manuscript presents the results of an in-depth study on new coumarin derivatives and their mechanism of action in neuroinflammation. Overall, I believe the quality of the text is more than sufficient for the paper to be published in a high- ranking magazine, such as Int J Mol Sci, and may be of reasonably high interest for the readers. I have to say that the level of English used throughout the manuscript is superb and enables the reader to follow the story without additional problem caused by so-often poor presentation.

Abstract gives all the necessary information about the contents of the paper, keywords are appropriately chosen and all the necessary literature is listed as required. I have noticed that approximately 10 references out of 54 are self-citations, but in my opinion, this is not a problem, since they correctly refer to your previous work in this field.

Answer: Dear Reviewer 1, many thanks for your positive feedback on our paper.

Introduction gives all the necessary information for the readers. It may be a bit longer and too detailed in some places, but I do not insist on its shortening. There is only one detail that I believe would make the presentation of the whole paper even more “reader friendly”. As a medicinal chemist I am very fond of structures and the mere description of KITs and other compounds in Lines 102–159 gives little or almost no information on their structure. For old compounds, the reader may browse through the previous papers, but the process ids time consuming. Since KIT C and KIT H are novel molecules, I suppose their structures have not been disclosed yet. Figure with their structures is mandatory, I would suggest that chemical structures of KIT 10, KIT 17, O-1602 and ML 193 are also included.

Answer: Thanks a lot for this good comment, we added the chemical structures of KIT C, KIT H, O- 1602 and ML 193 (line 567 to 574).

Materials and methods section gives sufficiently data on research work and is well presented. The methodology of the research is sound. Results are clearly presented and supported with large number of Figures. The same is true with Discussion. All parts are written in such a detail, that I have no further suggestions.

Reviewer 2 Report

The manuscript entitled “Functional selectivity of coumarin derivates acting via GPR55 in neuroinflammation” by Apweiler et al. describes some functional studies using two coumarin-based compounds, KIT C and KIT H, in human neuroblastoma cells and murine primary microglia cultures. It seems that this manuscript is a member of a series of studies using similar experimental setup and rationale (doi:10.3390/ijms22052503, doi:10.3390/ijms222111665 and doi.10.3389/fphar.2021.789074). Although the work reported here is interesting, properly executed and generally well-written, it needs further background information in the Introduction section, refinements and clarifications in the Methods section, and a more streamlined and topic-oriented Discussion. The use of further references, where indicated, would be an improvement. A major revision is necessary.

Major points:

1) The authors report that the drugs KIT C and KIT H were used in 0.1–25 microM final concentrations (100 nM-25,000 nM). This concentration range is typically too high for specific receptorial interaction (where a typical KD value would be in the nanomolar range). Moreover, O-1602, an agonist, and ML 193, an antagonist, have similarly high final concentrations when tested (5 microM and 25 microM, respectively). It seems that this receptor has a rather low affinity for the compounds used. Aspecific intermolecular interactions at these large concentrations cannot be predicted or avoided. Please comment on this (with references) in the Discussion section.

2) Even though this work is not a receptor binding study, the readers should be informed about the known pharmacological characteristics of GPR55, the subject of this study. Are there any receptor binding data (KD and Bmax values of KIT C and KIT H, and IC50 values in inhibition studies for O-1602 and ML 193) available to evaluate the real affinity, specificity, and selectivity, etc., of these compounds for GPR55? If yes, please explain (with references) in the revised manuscript.

3) There are certain similarities between this manuscript and a very recent one by the same group published in IJMS (doi.org/10.3390/ijms222111665). Both studies start with the description of KIT C and KIT H on cell viability. Figure 1 in this manuscript and Figure 1 in “Targeting Oxidative Stress: Novel Coumarin-Based Inverse Agonists of GPR55” by Apweiler et al. (Published: 28 October 2021) is essentially the same except for the inclusion of data from the use of O-1602 and ML 193. This similarity should be mentioned/referenced in the revised manuscript.

4) What is the rationale for using both a human neuroblastoma cell line (transformed, neural crest-derived cells that are capable of unlimited proliferation) and mouse primary microglial cells? What does one type of cell culture have that the other doesn’t have? Please explain. What is the biological significance that only KIT C decreased PGE2-levels in primary microglia and not in neuroblastoma cells?

5) A comparison of GPR55 characteristics, and its intracellular signaling pathway(s), in these cell lines should be included (with references) in the Discussion section.

6) There are nice dose-response relationships throughout the manuscript (Figures 1, 2, 4, 8). They are strikingly lacking for Figures 5, 6, 7; among these, Figures 4B and 5B are very interesting as they do seem to contradict each other, especially for KIT C (see also Minor point No. 10 below). The authors should elaborate further on this in the revised manuscript.

7) Line 558: “... to obtain pure microglial cell cultures.” The composition of the cultured cells in the 6-well plates should be determined using multicolor fluorescent immunocytochemistry/confocal microscopy using cell-specific markers. Please provide proof of the purity of microglial cultures using microglial cell-specific markers. The cited reference for this section (Saliba et al., 2017) did not check for and did not report proof of microglial purity. As contaminating cell types could compromise the data or the interpretation of it as reported in the manuscript, this is a critical point.

Minor points:

8) Lines 29-32: The authors state “Novel coumarin derivates have been shown to elicit anti-neuroinflammatory effects via G-protein coupled receptor (GPR)55, with possibly reduced side-effects compared to the known anti-inflammatory drugs.” Unfortunately, it is only speculation as the authors do not provide examples of any side effects for KIT C and H. It is difficult to assess their possible superiority over other (which one?) anti-inflammatory drugs without knowing their own side effects...

9) Lines 366-374: This part of the Discussion deals with negative results; moreover, “data not shown” was evoked 3 times. Taken together, these few lines are not convincing and do not form an important part of the narrative; it should probably be re-written.

10) The possible reason for the difference between mRNA levels and protein levels of COX-2 (see Lines 398–401, and also cited Ref. 45) is a bit confusing, especially the “can be affected by G-protein coupled receptors” part. Were they activated concomitantly with the activation of GPR55 by KITs? The authors should be more specific even when theorizing.

11) How many human neuroblastoma cells were cultured, passaged, used, etc. (line 537)?

12) How many cells were seeded for the generation of primary mouse microglia cultures (line 553)? How many “floating” microglial cells were collected at DIV12 (line 557)? Was it a spontaneous affair? No shaking was involved? The authors claim that the cells were collected at subDIV1. What was the cell count and the survival rate at this time? (The cells were used within a day after subculturing; the medium was changed and the cells were used after 1 h; quite a busy day for those cells.)

Author Response

The manuscript entitled “Functional selectivity of coumarin derivates acting via GPR55 in neuroinflammation” by Apweiler et al. describes some functional studies using two coumarin-based compounds, KIT C and KIT H, in human neuroblastoma cells and murine primary microglia cultures. It seems that this manuscript is a member of a series of studies using similar experimental setup and rationale (doi:10.3390/ijms22052503, doi:10.3390/ijms222111665 and doi.10.3389/fphar.2021.789074). Although the work reported here is interesting, properly executed and generally well-written, it needs further background information in the Introduction section, refinements and clarifications in the Methods section, and a more streamlined and topic-oriented Discussion. The use of further references, where indicated, would be an improvement. A major revision is necessary.

Answer: Dear Reviewer 2, many thanks for taking the time and reviewing our manuscript.

Major points:

1) The authors report that the drugs KIT C and KIT H were used in 0.1–25 microM final concentrations (100 nM-25,000 nM). This concentration range is typically too high for specific receptorial interaction (where a typical KD value would be in the nanomolar range). Moreover, O-1602, an agonist, and ML 193, an antagonist, have similarly high final concentrations when tested (5 microM and 25 microM, respectively). It seems that this receptor has a rather low affinity for the compounds used. Aspecific intermolecular interactions at these large concentrations cannot be predicted or avoided. Please comment on this (with references) in the Discussion section.

Answer: We have discussed the used concentrations as well as the possibility of non-specific intermolecular interactions in the revised manuscript (line 503 to 526, see answer to point two as well). However, the determination of markers (KD, IC50 etc.) of receptor affinities is usually executed in artificial systems, for example in in silico or membrane preparation models. In cellular systems, higher concentrations are necessary for this reason, because only a small amount of compound is actually available at the receptors and the cells do not overexpress the receptor GPR55. The GPR55 specific activity assay showing comparable doses to activate GPR55 suggests that the effects of KIT C and KIT H on COX-2/PGE2 are mediated via GPR55.

2) Even though this work is not a receptor binding study, the readers should be informed about the known pharmacological characteristics of GPR55, the subject of this study. Are there any receptor binding data (KD and Bmax values of KIT C and KIT H, and IC50 values in inhibition studies for O-1602 and ML 193) available to evaluate the real affinity, specificity, and selectivity, etc., of these compounds for GPR55? If yes, please explain (with references) in the revised manuscript.

Answer: Thanks a lot for this good comment. We have added EC50 for O-1602 and IC50 for ML 193 as well as for two related coumarin-derivates KIT 10 and KIT 17 in the Discussion. For KIT C and KIT H, the IC50 is unknown. Moreover, Bmax is not published for the GPR55 yet and KD is not available for the examined GPR55 ligands. We have combined points 1 and 2 in the Discussion since they connect to each other (line 503 to 526). The IC50 data of KIT 10 and Kit 17 were determined using a cellular system and are therefore in μM range when compared to ML 193.

3) There are certain similarities between this manuscript and a very recent one by the same group published in IJMS (doi.org/10.3390/ijms222111665). Both studies start with the description of KIT C and KIT H on cell viability. Figure 1 in this manuscript and Figure 1 in “Targeting Oxidative Stress: Novel Coumarin-Based Inverse Agonists of GPR55” by Apweiler et al. (Published: 28 October 2021) is essentially the same except for the inclusion of data from the use of O-1602 and ML 193. This similarity should be mentioned/referenced in the revised manuscript.

Answer: We included a reference to our previous paper in the revised version of the manuscript (line 163+164, line 176+177).

4) What is the rationale for using both a human neuroblastoma cell line (transformed, neural crest- derived cells that are capable of unlimited proliferation) and mouse primary microglial cells? What does one type of cell culture have that the other doesn’t have? Please explain. What is the biological significance that only KIT C decreased PGE2-levels in primary microglia and not in neuroblastoma cells?

Answer: This is a good point raised by the reviewer. We started to examine effects of KIT C and KIT H on neurons and neuronal cells, since neurons are connected to various pathological conditions in neuropsychological diseases. They are affected by neuroinflammatory processes, that often rely on activated microglia cells e.g. releasing pro-inflammatory mediators explaining the use of primary microglial cells in our manuscript. Therefore, different role and interaction of both cell types are of importance in the emergence and maintenance of neuroinflammation as well as in the symptomatic appearance of diseases, such as depression, schizophrenia, and Alzheimer’s Disease. The different effect of KIT H on PGE2 in SK-N-SH-cells and in primary microglia support the hypothesis that the GPR55 might show different selectivity to ligands between species (human vs. mice). Another explanation for the observed effect differences of KIT H might be a varying effective profile between different cell types (neuroblastoma vs. microglia). The receptor density and the coupled pathways might differ dependent on the cell types as described before, and KIT C and KIT H might enfold different effects in microglia due to their chemical structure.

5) A comparison of GPR55 characteristics, and its intracellular signaling pathway(s), in these cell lines should be included (with references) in the Discussion section.

Answer: Differences between human (SK-N-SH-cells) and mouse (primary microglia) in GPR55 characteristics regarding its mRNA and aminoacidic structure of the GPR55 protein have been discussed against the backdrop of species-selective effects of the investigated coumarin-based compounds (starting at line 482). The GPR55 is connected to a lot of different pathways, therefore we focused on the most relevant ones for COX-2 regulation and consecutive PGE2-synthesis. These are the MAP-kinases (p38 MAPK, Erk 1/2, SAPK/Jnk), NFAT, NF-kB (starting at line 363). There is no evidence, that the signaling pathways coupled to the GPR55 might differ between mouse and human, so regarding our results we can’t further elaborate pathways of interest or characterization of GPR55. We did not find any publications focusing on differences between GPR55 signaling pathways in mice and humans.

6) There are nice dose-response relationships throughout the manuscript (Figures 1, 2, 4, 8). They are strikingly lacking for Figures 5, 6, 7; among these, Figures 4B and 5B are very interesting as they do seem to contradict each other, especially for KIT C (see also Minor point No. 10 below). The authors should elaborate further on this in the revised manuscript.

Answer: The revised manuscript contains an elaboration regarding Figures 4B and 5B (line 418-425), that only contradict each other on the first view. The enzymatic activity of COX-2 is measured after 30 minutes of pretreatment with KIT C or KIT H but changes in protein levels cannot be found earlier than 8 hours of treatment in Western blot. Therefore, the investigated coumarin derivates do not specifically inhibit the COX-2 enzymatic activity like COX-2 inhibitors but rather achieve the PGE2 reduction by GPR55-pathway dependent COX-2 synthesis inhibition.

7) Line 558: “... to obtain pure microglial cell cultures.” The composition of the cultured cells in the 6- well plates should be determined using multicolor fluorescent immunocytochemistry/confocal microscopy using cell-specific markers. Please provide proof of the purity of microglial cultures using microglial cell-specific markers. The cited reference for this section (Saliba et al., 2017) did not check for and did not report proof of microglial purity. As contaminating cell types could compromise the data or the interpretation of it as reported in the manuscript, this is a critical point.

Answer: Thank you for this good comment. The protocol is based on the publication of Gebicke- Haerter et al. 1989 with the microglial cell culture protocol established in our laboratory by him (10.1523/JNEUROSCI.09-01-00183.1989), that demonstrated a purity of >98% using LPS-free growing medium. We have added additional information (line 600 to 614) on purity of the microglial cultures shown in this previous publication and added the reference in line 598 as well.

Minor points:

8) Lines 29-32: The authors state “Novel coumarin derivates have been shown to elicit anti- neuroinflammatory effects via G-protein coupled receptor (GPR)55, with possibly reduced side-effects compared to the known anti-inflammatory drugs.” Unfortunately, it is only speculation as the authors do not provide examples of any side effects for KIT C and H. It is difficult to assess their possible superiority over other (which one?) anti-inflammatory drugs without knowing their own side effects...

Answer: Thanks for raising this import point. Coumarin drugs are already used in medicine, for example as vitamin K antagonists and their side-effects are milder compared to glucocorticoids for example. An important possible side effect of coumarin is hepatotoxicity (10.1136/pgmj.71.841.701-a), warfarin is known for bleeding side-effects for example. When compared to glucocorticoids with a broad range of side effects or NSAIDs, that can cause serious nephrological and gastroenteric side effects as well, the use of coumarin derivates is possibly beneficial for at least some patients, for instance with a certain risk profile. The used concentrations of KIT C and KIT H are comparable to warfarin blood concentrations used for anticoagulation therapy (10.1016/j.jyp.2013.02.001). As in drug registration studies, we focused on biological effectiveness of both compounds on neuroinflammation first. We agree that the side effects should be object of further studies. Due to the limited word count, we did not add additional information to the abstract.

9) Lines 366-374: This part of the Discussion deals with negative results; moreover, “data not shown” was evoked 3 times. Taken together, these few lines are not convincing and do not form an important part of the narrative; it should probably be re-written.

Answer: We have revised the mentioned part of the Discussion and have shortened it (368 to 374). However, we think it is important to mention the examined pathways, even if those were not affected by KIT C and KIT H. In our opinion, it is important to exclude especially MAPK-pathways and NF-kB involvement in the observed PGE2-reduction since those pathways are coupled to GPR55 and associated with COX-2 enzyme levels.

10) The possible reason for the difference between mRNA levels and protein levels of COX-2 (see Lines 398–401, and also cited Ref. 45) is a bit confusing, especially the “can be affected by G-protein coupled receptors” part. Were they activated concomitantly with the activation of GPR55 by KITs? The authors should be more specific even when theorizing.

Answer: We tried to be more specific and clarify this passage in the revised manuscript (line 401 to 406).

11) How many human neuroblastoma cells were cultured, passaged, used, etc. (line 537)?

Answer: We added the information about cell number/density in the SK-N-SH cell part (line 577, 582 to 588).

12) How many cells were seeded for the generation of primary mouse microglia cultures (line 553)? How many “floating” microglial cells were collected at DIV12 (line 557)? Was it a spontaneous affair? No shaking was involved? The authors claim that the cells were collected at subDIV1. What was the cell count and the survival rate at this time? (The cells were used within a day after subculturing; the medium was changed and the cells were used after 1 h; quite a busy day for those cells.)

Answer: We have clarified the part of the Method Section 4.3.2 Primary mouse microglia cultures and added further information about cell counts and the process of preparing the cultures (line 600 to 614).

Round 2

Reviewer 2 Report

The revised manuscript entitled “Functional selectivity of coumarin derivates acting via GPR55 in neuroinflammation” by Apweiler et al. deals diligently with the criticisms and suggestions raised by this reviewer. All my original major and minor points (12 in total) were properly addressed. Further discussions regarding the suggestions were added as requested. These additions fit the narrative of the manuscript and are properly referenced. There are nine new references in the manuscript that support the information added mainly to the Discussion and the Materials and Methods sections. The revised manuscript improved significantly and can be accepted for publication in the present form.